# Ruthenium–Thymine Acetate Binding Modes: Experimental and Theoretical Studies

**Silvia Bordoni** [1,*] , **Stefano Cerini** [1] , **Riccardo Tarroni** [1] , **Magda Monari** [2] , **Gabriele Micheletti** [1] and **Carla Boga** [1]

[1] Dipartimento di Chimica Industriale "Toso Montanari", Università di Bologna, Viale Risorgimento 4, 40136 Bologna, Italy; stefano.cerini3@unibo.it (S.C.); riccardo.tarroni@unibo.it (R.T.); gabriele.micheletti3@unibo.it (G.M.); carla.boga@unibo.it (C.B.)

[2] Dipartimento di Chimica "Giacomo Ciamician", Università di Bologna, Via Selmi 2, 40126 Bologna, Italy; magda.monari@unibo.it

[*] Correspondence: silvia.bordoni@unibo.it

**Abstract:** Ruthenium complexes have proved to exhibit antineoplastic activity, related to the interaction of the metal ion with DNA. In this context, synthetic and theoretical studies on ruthenium binding modes of thymine acetate (THAc) have been focused to shed light on the structure-activity relationship. This report deals with the reaction between dihydride ruthenium mer-[Ru(H)$_2$(CO)(PPh$_3$)$_3$], **1** and the thymine acetic acid (THAcOH) selected as model for nucleobase derivatives. The reaction in refluxing toluene between **1** and THAcOH excess, by H$_2$ release affords the double coordinating species κ$^1$-(O)THAc-, κ$^2$-(O,O)THAc-[Ru(CO)(PPh$_3$)$_2$], **2**. The X-ray crystal structure confirms a simultaneous monohapto, dihapto- THAc coordination in a reciprocal facial disposition. Stepwise additions of THAcOH allowed to intercept the monohapto mer-κ$^1$(O)THAc-Ru(CO)H(PPh$_3$)$_3$] **3** and dihapto *trans*$_{(P,P)}$-κ$^2$(O,O)THAc-[Ru(CO)H(PPh$_3$)$_2$] **4** species. Nuclear magnetic resonance (NMR) studies, associated with DFT (Density Function Theory)-calculations energies and analogous reactions with acetic acid, supported the proposed reaction path. As evidenced by the crystal supramolecular hydrogen-binding packing and $^1$H NMR spectra, metal coordination seems to play a pivotal role in stabilizing the minor [(N=C(OH)] lactim tautomers, which may promote mismatching to DNA nucleobase pairs as a clue for its anticancer activity.

**Keywords:** ruthenium; thymine acetate; X-ray crystal structure; DFT-energy calculations; *cis-trans* interconversion; H-bonding intermolecular network

## 1. Introduction

In the last decade, a plethora of novel developments on ruthenium complexes exhibited a remarkable efficiency as potential metal-drugs in anticancer treatments together with a consistent reduction of the related toxic issues exhibited by the former Pt-agents. The ruthenium complexes possessing anticancer activity exhibit a plethora of different structures, mainly belonging to coordination chemistry with labile chloride groups [Ru(II)(biphenyl)Cl(en)]$^+$, where en = 1,2-ethylenediamine [1] or the efficient solvato-complexes [Ru(III)Cl$_4$(DMSO)(imid)][imidH] [imid=imidazole] [2]; at the same time, organometallic piano stool compounds bearing the ubiquitous p-cymene [Ru(p-cymene)Cl(H2O) (PTA)]$^+$, arene or cyclopentadienyl groups and a variety of chelate donor ligands as bipyridine moieties as shown by [Ru(η$^5$-C$_5$H$_5$)(PPh$_3$)bipy)][CF$_3$SO$_3$][bipy=bypyridyl][3]. Therefore, it is of fundamental importance to disclose more evidence on the structure–effect relationship. In this respect, great efforts have been made in ruthenium novel drugs [1–16] exploration, focused on the interaction modes towards DNA nucleic bases [17–23]. A useful approach to shed some light on the mechanism related to biological action effectiveness [24] is to study the coordination modes of a metal-anchored thymine acetic acid (THAcOH), proposed here as ligand to design potential bioactive ruthenium complexes. Due to chelate properties,

the side arm carboxylate group may induce further stability, analogously to transferrin behavior in the inter-strand crosslinks. The reaction course will be explored by spectroscopic and structural investigations, DFT-calculated energy profiles and by comparing analogous reactivity with acetic acid. The intermolecular H-binding network in THAc-Ru systems, which is remarkably evident in the X-ray structure, prompt us to investigate the occurrence in solution. Evaluation of the [1]H nuclear magnetic resonance (NMR) intermolecular H-bonding [25] signals has been suggested by the help of DFT-calculated energies, through simulation of the interactions of two THAc⁻ anions, selected as model. By dealing with the mutagenic properties of metallodrugs in anticancer activity, Lippert's pioneering work [21–23,26–30] suggested that proton relocation, obtained by acid treatment, deprotonation, or H-bond interactions is responsible for the stabilization of rare tautomeric forms. The proton relocation can be a crucial key in mismatching DNA nucleobase pairs, by impeding cell replication.

## 2. Materials and Methods

### 2.1. General

All reactions were routinely carried out under argon atmosphere, using standard Schlenk techniques. Solvents were distilled immediately before use under nitrogen from appropriate drying agents. Chromatography separations were carried out on columns of dried celite. Glassware was oven-dried before use. Infrared spectra were recorded at 298 K on a PerkinElmer Spectrum 2000 FT-IR (Fourier transform infrared) spectrophotometer (Waltham, MA, USA), electrospray ionization mass spectrometry (ESI MS) spectra were recorded on a Waters Micromass ZQ 4000 (Milford, MA, USA), with samples dissolved in $CH_3OH$. All deuterated solvents were degassed before use. All NMR measurements were performed on a Mercury Plus 400 instrument (Oxford Instruments, Abingdon-on-Thames, UK). The chemical shifts for [1]H and [13]C were referenced to internal TMS. [1]H, [13]C correlation measured using gs-HSQC (gradient selected Heteronuclear Single Quantum Coherence) and gs-HMBC (gradient selected Heteronuclear Multiple Bond Correlation) experiments. All NMR spectra were recorded at 298 K. All the reagents were commercial products (Aldrich) of the highest purity available and used as received. $[RuCl_3 \cdot xH_2O]$ was purchased from Strem and used as received. Compound $[Ru(H)_2(CO)(PPh_3)_3]$ (**1**) was prepared by published methods [31].

### 2.2. Experimental Procedure

2.2.1. Synthesis of Complex. $\kappa^1 O$)-THA $\kappa^2(O,O)$-THAc-[Ru(CO)(PPh₃)₂], **2**

To a 100 mg of **1** were added upon stirring in refluxing toluene two equivalents of THAcH (40 mg, 0.218 mmol) for 240 min. After 15 min evolution of a gas was observed until the Ru-CO absorption at 1940 cm$^{-1}$ disappeared. After drying under vacuum, the light brown solid was washed with Et₂O (3 × 10 mL) to remove the released PPh₃. The crystallization occurred in CDCl₃ gave white crystals of **2** (90%, 97 mg). The same reaction was carried out by adding an excess of THAcH or two equivalents consecutively, until the starting material infrared (IR) absorption bands disappeared.

[$RuC_{53}H_{44}O_9P_2$; 988.15 g mol$^{-1}$]; IR (KBr, cm$^{-1}$) ***trans*-2** ν: Ru-CO 1969 vs; $\kappa^2$_THAc, $\nu_{asym}$ *C(O)O* 1653s, $\nu_{sym}$ *C(O)O* 1630s; $\kappa^2$_THAc ν *C(O)O* 1521 vs. $\kappa^1$_THAc ν *C(O)O* 1361 vs, (Δν = 280 cm$^{-1}$); [1]H-NMR (400, CDCl₃): δ 1.74 (s, 6H, Me $\kappa^1$_THAc + Me $\kappa^2$_THAc), 1.82 (aq, $\sum^2 J_{HP}$ = 13 Hz $CH_\alpha H_\beta$, $\kappa^1$_THAc); 4.29 (aq, $^5 J_{HP}$ = 89 Hz, $^2 J_{HH}$ = 17 Hz, $CH_\alpha H_\beta$, $\kappa^2$_THAc [C(O)O$\kappa^1$_THAc]; 5.85 (s, 1H,CH-methyne); 6.80–7.70 (m, 30H, 2 PPh₃); 7.90 (s, br,1H, NH), 8.10 (s, br, 1H, NH). [13]C{[1]H}(100 MHz, CDCl₃) δ 205.6 (t, CO $^2 J_{CP}$ = 15Hz,); 177.4 (COO THAc); 164.6 150.5 (CO, THAc); 141.7 (Cq THAc); 126–138 (PPh₃); 109.7 (CH); 49.5 (CH₂); 14.7 (Me) ***cis*-2** ν: Ru-CO 1969 vs; $\kappa^2$_THAc, $\nu_{asym}$ *C(O)O* 1658s, $\nu_{sym}$ *C(O)O* 1635s; $\kappa^2$_THAc ν *C(O)O* 1536 vs (Δν = 111 cm$^{-1}$) $\kappa^1$_THAc ν *C(O)O* 1358m, (Δν = 289 cm$^{-1}$); [1]H-NMR (400, CDCl₃): δ 1.05 (s, 6H, Me $\kappa^1$_THAc + Me $\kappa^2$_THAc), 1.43 (s, 2H, CH₂N); 3.27 (aq, $^5 J_{HP}$ = 828 Hz, $^2 J_{HH}$ = 16 Hz, $CH_\alpha H_\beta$, $\kappa^2$_THAc [C(O)O$\kappa^1$_THAc]; 4.43 (s, 1H, CH-methyne); 6.80–7.70 (m, 30H, 2 PPh₃); 7.67 (m, 2H, 2NH). The $\kappa^1$- and $\kappa^2$- coordination are undistin-

guishable because of the mutual scrambling at room temperature that affect either $^1$H or $^{13}$C NMR spectra. $^{31}$P{$^1$H}-NMR (161.9 MHz, CDCl$_3$): *cis* 57.0 (dd, $^2$J$_{Pa-Pb}$ = 15 Hz, 2P *cis*). 177,168 (CO$_{TH}$)$_{TRANS}$ 170, 173 (CO$_{TH}$)$_{CIS}$ 163 (COO)$_{TRANS}$, 162 (COO)$_{CIS}$, 150, 141, 125, 112 (J$^2_{CP}$ = 4 Hz) PPh$_3$ 109.5 (CH) $_{TRANS}$ 109 (CH) $_{CIS}$ 110 (CMe)$_{TRANS}$ 49 (broad NCH$_2$)$_{TRANS}$ 52 (broad NCH2)$_{CIS}$ 39 (Me)$_{TRANS}$ 32 (Me)$_{CIS}$.

### 2.2.2. Synthesis of Complexes *mer*–κ$^1$(O)THAc-[RuH(CO)(PPh$_3$)$_3$] (3a, 3b) + *trans*$_{(P,P)}$-κ$^2$(O,O)THAc -[RuH(CO)(PPh$_3$)$_2$], 4

To a 100 mg of 1 was added upon stirring in refluxing toluene an equimolar amount of THAcH (20 mg, 0.109 mmol). After 10 min evolution of a gas was observed until the Ru-CO absorption at 1940 cm$^{-1}$ disappeared. After 30 min the solid was dried under vacuum, the light brown solid was washed with Etp (3 × 10 mL) to remove the released PPh$_3$ and chromatographed on a celite column (7 × 1.5 cm). By elution with neat Et$_2$O a first pale green fraction was collected giving the monohapto acetate derivatives **3** (72 mg, 60%), whereas the elution with CH$_2$Cl$_2$ gave a purple fraction prevalently constituted by the dihapto species **4** (27 mg, 30%).

By changing the reaction times to 60 min the major fraction mainly afforded the purple fraction constituted by the chelate derivative **4** (78 mg, 85 %); ***mer*–κ$^1$(O)THAc-[Ru (CO)H(PPh$_3$)$_3$] (3a,3b)**, green powdered solid; IR (KBr, cm$^{-1}$) ν: Ru-CO 1918 vs; ν$_{as}$ C(O)O κ$^1_{THAc}$ 1683, ν$_s$ C(O)Oκ$^1_{THAc}$ 1654s (Δν = 175 cm$^{-1}$), 1480 m (P-C) 1434 vs. (P-C); 1361m C(O)O; $^1$H-NMR (400.0 MHz, CDCl$_3$: δ 6.8–7.7 (m, 45H, 3 PPh$_3$); 5.7 (s br, 1H, CH THAc), 5.4 (s br, 1H, CH THAc); 3.9 (s br, 2H, CH$_2$); 3.1 (s br, 2H, CH$_2$); 1.6 (s, 3H, Me); −6.1 (dt, J$_{P-H}$ = 112 Hz, J$_{P-H}$ = 24Hz, 1H); −7.2 (dt, J$_{P-H}$ = 112Hz, J$_{P-H}$ = 24 Hz, 1H); $^{13}$C{$^1$H}-NMR (100.6 MHz, CDCl$_3$: δ 205.6 (s br, CO); 203.9 (s br, CO); 177.3 (COO); 170.9 (COO); 167.9 (CO(1) THAc); 164.7 (CO(1) THAc); 151.5 (CO(2) THAc); 150.5 (CO(2) THAc); 142.8 (Cq THAc); 141.6 (Cq THAc); 135.3 (t, J$_{CP}$ = 5 Hz, PPh$_3$); 134.5 (t, J$_{PC}$ = 22 Hz, Cq PPh$_3$); 130.7 (s, CH$_{para}$ PPh$_3$); 128.0 (t, J$_{PC}$ = 4 Hz, CH PPh$_3$); 108.9 (CH THAc); 108.5 (CH THAc); 50.5 (CH$_2$); 49.5 (CH$_2$); 12.8 (Me); 12.7 (Me); $^{13}$C{$^1$H}-NMR (100.6 MHz, CDCl$_3$): δ $^{31}$P{$^1$H}-NMR (161.9 MHz, CDCl$_3$: δ 41.2 (s br, 1P); 36.5 (s br, 2P).

***trans*$_{(P,P)}$-κ$^2$(O,O)-[Ru(CO)H(PPh$_3$)$_2$THAc], 4** waxy reddish solid; IR (KBr, cm$^{-1}$) ν 1963 vs. (RuCO); 1683 vs, b; 1443, 1434 (P-C); $^1$H-NMR (400,0MHz,CDCl$_3$: δ 6.8–7.7 (m, 30H, 2 PPh$_3$); 5.4 (s br, 1H, CH THAc); 3.9 (s br, 2H, CH$_2$); 1.6 (s, 3H, Me); −17.0, (t, $^2$J$_{HP}$ = 20.0 Hz);$^{13}$C{$^1$H}-NMR (100.6 MHz: δ Ru-CO not observed, 179.8 (s, COO κ$^2$-THAc); 163.9 (s, CO(1) κ$^2$-THAc); 149.8 (s, CO(2) κ$^2$-THAc); 141.4 (s, Cq κ$^2$-THAc); 124.0–136.0 (PPh$_3$) 112.0 (s, CH κ$^2$-THAc); 64.4 (s, CH$_2$ κ$^2$-THAc); 13.4 (s, Me κ$^2$-THAc); $^{31}$P{$^1$H}-NMR (161.9 MHz, CDCl$_3$: δ 38.7 (d, $\sum$$^2$J$_{PH}$ = 14 Hz, 2P); Ms-ESI (MeOH, *m/z*): 1043 {20%, M=[Ru(CO)(THAc)$_2$(PPh$_3$)$_2$]$^+$+Na} 859 {25%, [Ru(CO)(THAc)(PPh$_3$)$_2$]$^+$+Na} 837 {100%, [Ru(CO)(THAc)(PPh$_3$)$_2$]$^+$+H}; 653 {15%, [Ru(CO)(PPH$_3$)$_2$]$^+$+H}.

### 2.3. Reactions of *1* with Acetic Acid

For optimizing the sake of product selectivity, we run the reactions by refluxing in different solvents and by using different quantities of reactant (Table 1).

**Table 1.** Selected conditions for the reaction of 1 with AcOH.

| Solvent, rfx | Ru/AcH Ratio | Time (min) | Species | Yield (%) |
|---|---|---|---|---|
| CHCl$_3$ | 1:2 | 240 | **7** | 65 |
| CDCl$_3$ r.t. | 1:1 | 14 h (r.t); | **5 + 6** | 50 (**6**) + 38 (**5**) |
| CDCl$_3$ | 1:2 | 30 | **7** | 50 (**7**) |
| CH$_2$Cl$_2$ | 1:2 | 40 | **5** | 70 |
| toluene | 1:1 | 240 | **5 + 6** | 70 (**5**) + 30 (**6**) |
| CPME | 1:2 | 40 | **7** | 70 (**7**) |

A 200 mg of 1 has added upon stirring in refluxing solvents with variable quantities of acetic acid (25 μL ~2 eq). After 5 min, evolution of a gas was observed until the Ru-CO

absorption at 1940 cm$^{-1}$ disappeared. After the reaction times displayed in Table 1 the solid was dried under vacuum and the light brown solid was washed with Etp (2 × 10 mL) to remove the released PPh$_3$ and filtered on a celite column (8 × 1.5 cm). By elution with neat Et$_2$O two fractions were collected giving a mixture of acetate adducts with prevalence of the desired product. Table 1 summarizes the reaction conditions of different reactions run by seeking the suitable conditions to reach the maximum conversion and selectivity.

*mer*-**κ$^1$(O)Ac-[Ru(CO)H(PPh$_3$)$_3$] 5** pale green oily solid; IR (KBr, cm$^{-1}$) ν: 1923 vs. (CO); $^1$H-NMR (400.0 MHz, CDCl$_3$): δ 6.80–780 (m, 45H; PPh$_3$); 1.28 (s, br 3H, CH$_3$COO); −7.13 (dt, $^2$J$_{PPcis}$ = 24 Hz, $^2$J$_{PPtrans}$ = 105 Hz, Ru-H); $^{13}$C{$^1$H}-NMR (100,6MHz): δ Ru-CO 204.3 (t, $^2$J$_{PC}$ = 13.9 Hz, CO$_{trans}$), 203.0 (dd, ∑S$^2$J$_{PaC}$ = 15.2 Hz, CO$_{cis}$), 189.3 (COO), 24.6 (Me); $^{31}$P{$^1$H}-NMR (161.9 MHz, CDCl$_3$): δ 36.3.

*trans$_{(P,P)}$*-**κ$^2$(O,O)Ac-[Ru(CO)H(PPh$_3$)$_2$] 6**, deep purple waxy solid; IR (KBr, cm$^{-1}$) ν: 1929 vs. (CO); $^1$H-NMR (400.0 MHz, CDCl$_3$): δ 6.80–780 (m, 30H; PPh$_3$); 0.62 (s, br 3H, CH$_3$COO);-16.50 ppm ($^2$J$_{HP}$ = 20 Hz); $^{13}$C{$^1$H}-NMR (100,6MHz): δ Ru-CO, 186.1 (COO), 135.5–126.9 (PPh$_3$); 30.3 (Me).$^{31}$P{$^1$H}-NMR (161.9 MHz, CDCl$_3$): δ 38.4.

*trans$_{(P,P)}$*-**κ$^1$(O)Ac,-κ$^2$(O,O)Ac-[Ru(CO)(PPh$_3$)$_2$] 7a,** pink solid; IR (KBr, cm$^{-1}$) ν: 1940 vs. (CO), **1625 COO, 1465 (κ2) 1434sy (κ1)** (COO); $^1$H-NMR (400.0 MHz, CDCl$_3$): δ 7.80–6.80 (m, 30H; PPh$_3$); 0.69 (s, br 6H, CH$_3$COO); $^{13}$C{$^1$H}-NMR (100,6 MHz): δ 206.0 (t, $^2$J$_{PC}$ = 14 Hz, CO$_{trans}$), 186.0 (COO), 133.5 (t, J$_{PC}$=8Hz, Cq PPh$_3$); 133.3 (t, J$_{PC}$ = 5 Hz, CH PPh$_3$); 129.5 (s, CH$_{para}$ PPh$_3$); 127.2 (t, $^2$J$_{PC}$ = 5 Hz, CH PPh$_3$); 21.9 (Me). $^{31}$P{$^1$H}-NMR (161,9MHz, CDCl$_3$): δ 39.1.

*cis$_{(P,P)}$*-**κ$^1$(O)Ac,-κ$^2$(O,O)Ac-[Ru(CO)(PPh$_3$)$_2$] 7b,** violet oily solid; IR (KBr, cm$^{-1}$) ν: 1943 vs. (CO), 1515, 1434 (COO); 1482, 1466 (P-C) $^1$H-NMR (400.0 MHz, CDCl$_3$): δ 7.84–6.95 (m, 30H; PPh$_3$); 0.52 (s, br 6H, CH$_3$COO); $^{13}$C{$^1$H}-NMR (100.6 MHz): 204.7 (dd, $^2$J$_{PaC}$ = 19 Hz, $^2$J$_{PbC}$ = 16 Hz CO$_{cis}$), 189.30, (COO), 189.27 (COO), 133.5 (t, J$_{PC}$ = 8 Hz, Cq PPh$_3$); 133.3 (t, J$_{PC}$ = 5 Hz, CH PPh$_3$); 129.5 (s, CH$_{para}$ PPh$_3$); 127.2 (t, $^2$J$_{PC}$ = 5 Hz, CH PPh$_3$); 22.6 (Me), 22.3 (Me). $^{31}$P{$^1$H}-NMR (161.9 MHz, CDCl$_3$): δ 35.5.

## 2.4. X-ray Crystallography

The X-ray intensity data were measured on a Bruker Apex II CCD diffractometer. Cell dimensions, and the orientation matrix was initially determined from a least-squares refinement on reflections measured in three sets of 20 exposures, collected in three different ω regions, and eventually refined against all data. A full sphere of reciprocal space was scanned by 0.3° ω steps. The software SMART was used for collecting frames of data, indexing reflections and determination of lattice parameters. The collected frames were then processed for integration by the SAINT program, and an empirical absorption correction was applied using SADABS. The structures were solved by direct methods (SIR 97) and subsequent Fourier syntheses and refined by full-matrix least-squares on F2 (SHELXTL) [32–35] using anisotropic thermal parameters for all non-hydrogen atoms. The structure contained significant void space (527 Å$^3$) and residual electron density that could not be meaningfully modelled; hence the SQUEEZE routine of PLATON was employed.

Cambridge Crystallographic Data Centre (CCDC)-1,868,289 contains the supplementary crystallographic data for this paper. These data can be obtained free of charge at www.ccdc.cam.ac.uk/conts/retrieving.html, accessed on 17 September 2018 (or from the Cambridge Crystallographic Data Centre, 12 Union Road, Cambridge CB2 1EZ, UK; Fax: +44-1223-336-033; e-mail: deposit@ccdc.cam.ac.uk).

## 2.5. Computational Details

DFT calculations were performed using the Molpro2010 software [36]. Due to the size of the involved species, geometry optimizations and free energy calculations (at 298 K) were undertaken with the small LANL2DZ [37] basis and the B3LYP [38] functional. Additional single point energy calculations with the larger def2-TZP [39] basis and the same B3LYP functional were performed at the previously optimized geometries.

The final energy of each structure, used to evaluate the relative free energies of the various products and intermediates, was built by summing the difference between the LANL2DZ electronic and free energies to the def2-TZP electronic energy. All calculations were performed *in vacuo*, i.e., commonly without considering the effect of the dielectric constant of the solvent unless otherwise stated.

## 3. Results

### 3.1. Reactions of [Ru(CO)H$_2$(PPh$_3$)$_3$] **1** with THAcOH

The Ru-hydride *mer*-[Ru(CO)H$_2$(PPh$_3$)$_3$] [40], **1** reacted in refluxing toluene within 240 min with thymine-acetic acid (THAcH) excess, by molecular hydrogen release until no [1]H NMR signals for residual Ru-hydride in the crude solution were observed. Attribution to a structure, showing IR frequency at 1970 cm$^{-1}$ for Ru-carbonyl function and simultaneous mono- and di-hapto-acetate-Ruthenium coordination like κ$^1$(O)-THAc, κ$^2$(O,O)-THAc[Ru(CO)(PPh$_3$)$_2$], **2** has been suggested on the basis of the IR frequency pattern. Namely, antisymmetric, and symmetric stretching vibrations of the COO$^-$ group of acetate at 1653 and 1363 (Δν = 290 cm$^{-1}$) indicate κ$^1$-THAc coordination, whereas narrower frequency differences (between ν = 1630 and 1532, Δν = 98 cm$^{-1}$) are typical for κ$^2$-THAc chelate mode. Species **2** in CDCl$_3$ showing a single sets of [1]H NMR resonances has been assigned to the less sterically hindered *trans*$_{P,P}$-configuration on the basis of stability considerations reported for similar complexes [41,42]. The [1]H NMR signals at δ 1.74, and 5.85 ppm, are respectively attributed to methyl and methyne thymine substituents, suggesting facile fluxional exchange at room temperature between κ$^1$- and κ$^2$-THAc rings, which may plausibly occur through a penta-coordinated transient species, in which the THAc ligands coordinatively exchange by a merry-go-round (THAc) mutual interconversion. Then, the *mer*$_{THAc^-}$ stereogeometry designation is independent from their hapticity mode. The [1]H NMR signal at δ 2.35 is associated to the κ$^2$-NCH$_2$ group in *trans*$_{P-P}$-*mer*$_{THAc}$-form, whereas a multiplet centered at 4.29 ppm showing a typical roof effect (doublet of doublet CH$_\alpha$H$_\beta$, $^2$J$_{HH}$ = 17; $^2$J$_{HH}$ = 90 Hz), is attributed to the NCH$_\alpha$H$_\beta$ methylene moiety of the κ$^1$-carboxy moiety. DFT molecular modelling calculations reveal that the Ru-OC(O) carboxy rotation is hampered by the axial phosphine steric crowding. By contrast, the absence of detectable J$_{H-H}$ coupling in the case of the dihapto-coordination may be attributed to the methylene group of chelate carboxy function, lying in planes that are mutually perpendicular (Karplus effect) [43]. This reciprocal disposition was evidenced both by the DFT *trans*$_{P-P}$- calculated isomer (94° in CDCl$_3$ solution) and by the X-ray crystal structure of the *cis* isomer (87° A) (Figure 1). DFT calculations predicted also a low energy interconversion process from *trans*$_{P,P}$-*mer*$_{THAc}$ to *cis*$_{P,P}$-*fac*$_{THAc}$ species (ΔE = 3.7 KJ mol$^{-1}$ in vacuo), which decreased to 0.8 kJ mol$^{-1}$, if the calculations take into account of the solvent effect (CDCl$_3$). We supposed that Ru-O cleavage of the chelate-THAc could promote PPh$_3$ migration from axial to equatorial position in a pentacoordinated intermediate species, as observed in the case of the precursor **1** solicited by light [44,45]. Accordingly, complete interconversion from *trans*$_{P,P}$-*mer*$_{THAc}$ to *cis*$_{P,P}$-*fac*$_{THAc}$ isomer occurs at r.t. in CDCl$_3$ within 3 h. Two [31]P NMR doublets centered at δ 46.5, δ 43.2; (J$_{PP}$ = 25 Hz) are representative for *cis-fac* **2** form, while a broad doublet detected at 35.9 ($^2$J$_{P,P}$ = 49 Hz) is the only observed residual signal due to the *trans-mer* form. The rapid mutual room temperature scrambling of the κ$^1$- and κ$^2$- coordinated THAc fragments made undistinguishable both the Me and the CH methyne signals in [1]H or [13]C NMR spectra. On the contrary, the THAc-methylene moiety and the phosphine [31]P NMR signals resulted both in being non-equivalent, since they were affected by punctual different molecular environments. No-planar disposition of the thymine rings altered the global symmetry, therefore was responsible for doublets related to [1]H NMR mutual coupling even in *trans*-configuration. As expected, the IR frequency pattern of the *cis* form was totally analogous to the *trans*-, but slightly lower-shifted. Due to the scarce solubility in less polar solvent, the [13]C NMR spectrum, which was recorded within 24 h for resolution requirements, exhibited a broad triplet signal at δ 205.6 for the Ru-CO of the former *trans*$_{P,P}$-*mer*$_{THAc}$ isomer next to a multiplet at δ 203.9 attributable to

the growing *cis$_{P,P}$-fac$_{THAc}$* species (Scheme 1). The $^{13}$C signals were attributed in a spectrum of *trans:cis* = 2:1 mixture, where the growing signals, related to the *cis* form, were adjacent to the *trans* one. Both the isomers were unstable in solution, and exhibited complete decomposition within 36 h at room temperature in polar coordinating solvents as DMSO (dimethylsulphoxide).

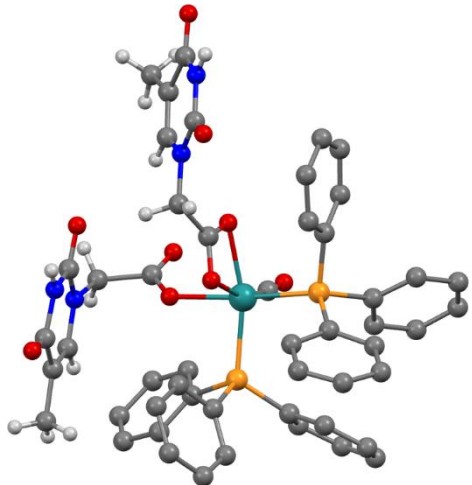

**Figure 1.** X-ray structure of κ$^1$(O)-THAc, κ$^2$(O,O)-THAc [Ru(CO)(PPh$_3$)$_2$] *cis*-**2**. Hydrogen atoms have been omitted for clarity.

**Scheme 1.** The fluxional interconversion between *trans*-, *mer*-**2** isomer and the related *cis,fac*-form envisages the formation of a transient penta-coordinated species, bearing double κ$^1$-THAc coordination mode which affords the complete transformation to the *cis*-**2** isomer.

The *cis$_{P,P}$-fac$_{THAc}$* configuration in CDCl$_3$ is calculated to be almost isoenergetic and stabilized by a multi-intermolecular H-binding network, which is clear in the X-ray crystal packing. In the crystal environment, the κ$^1$-THAc ligand is able to intermolecularly couple through Watson–Crick N-H···O interactions with the chelate κ$^2$ (O,O)-THAc counterpart of the adjacent molecule, forming a supramolecular metallacycle, reinforced by antiparallel thymine ring π–π stacking together with a phosphine phenyl C-H···O interaction [12]. In the crystal packing (Figure S2), symmetry related molecules of **2** are connected via two pairs of N-H···O hydrogen bonds between κ$^1$(O)THAc -and κ$^2$ (O,O)THAc ligands of neighboring molecules thus forming dimers centered about inversion centres. Further

stabilization of the dimer is achieved by π–π stacking of the thymine rings (centroid-centroid distance 3.65 Å) (Figure 2). In this way the thymine-1-acetic ligand preserves the N-H–O interactions observed in the crystal packing of thymine-1-acetic acid. A similar self-assembly showing intermolecular N-H–O hydrogen bonds and the π–π stacking of the thymine residue has been previously observed, for example, in the paddle-wheel dicopper complex $[Cu_2(O(O)CCH_2-T)_4(DMSO)_2]$. The supramolecular architecture is completed by an intricate intermolecular C-H–O network involving the hydrogen atoms of two phenyl rings belonging to two PPh$_3$ groups of two different molecules and the uncoordinated O atom (O6) of the monodentate THAc ligand. An analogous self-assembly interaction was observed in the THAcOH crystal structure [46–49]. Therefore, the *cis*-geometry of **2** could also be ascribed to H-bonding interactions between the THAc-rings and the low solubility exhibited by the *cis*-form attributed to the inter-molecular H-bonding network in the solid packing as inferred by the small energy gap (3.7 kJ mol$^{-1}$) between the *trans* and *cis* configurations, calculated by DFT. It is worth noting that the Ru-CO function lies on the same side (*syn*) of the C(O)O carboxy moiety, in accord with the DFT calculations, which designate this stereogeometry as more stable in the observed *cis*-fac and in the unstable *trans-mer* form (See Scheme 4 for calculations). The NMR spectra relative to *trans*-**2** and *cis*-**2** isomers both show the presence of many solvents, which have been used during the synthesis (toluene), in chromatographic processes (CH$_2$Cl$_2$ and Et$_2$O), or in multi-solvent stratification procedures for crystallization (CH$_2$Cl$_2$, Et$_2$O and Etp). Fluxional behaviour, that plausibly promotes interconversion from *trans* to *cis* forms, implies generation of pentacoordinated intermediates, which in turn are stabilized by coordinating solvents, as diethylic or methyl-cyclopentyl ether. As shown by $^1$H NMR spectra, species **2** strongly attract solvent molecules up to the second coordination sphere, exhibiting a solvatochromic effect in solution, and solvent inclusion in the crystals, selected for X-ray diffraction studies.

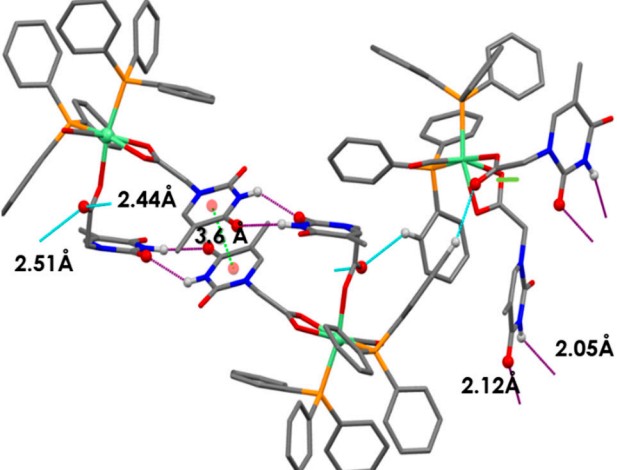

**Figure 2.** Distinct H-bond interactions have been evidenced by different color notifications. —— Expected intermolecular Watson-Crick H-interactions Å NH-OC between two adjacent THAc rings. —— OC(O)-H (C5H5-P) between the κ$^1$-THAc C(O)O and the proton belonging to a phenyl of PPh$_3$ of the adjacent molecule O6-H45 (2.51) O-H31 (2.44) N2H-O7 (2.12) N4H-O4 (2.05). - - - π–π stacking between two aromatic rings (3.6 Å) of THAc ring belonging to opposite molecules.

### 3.2. Studies on the Reaction Path: Isolation of the Intermediates mer–κ$^1$(O)-THAc [Ru(CO)H(PPh$_3$)$_3$] 3a, 3b and trans$_{(P,P)}$-[κ$^2$(O,O)-THAc-[Ru(CO)H(PPh$_3$)$_2$] 4

With the purpose to investigate the reaction course of **2**, a stepwise addition of two subsequent equivalents of THAcH intercepted two intermediates, which were spectroscopically characterized and analyzed by DFT calculations. The reaction (1:1 molar ratio) were stopped after 30 min. and the dried residue was eluted by Et$_2$O on a celite pad giving two distinct fractions. The nature of the more abundant pale green band (60%) has been assigned to a monohapto-coordinated *mer*–κ$^1$(O)THAc-[Ru(CO)H(PPh$_3$)$_3$] **3a,**

**3b** (Scheme 1) on the basis of a single Ru-CO IR absorption at ν = 1918 cm$^{-1}$ and the characteristic acetate band pattern. The $^{31}$P{$^1$H} NMR broad signals at δ 41.2 and 36.5 in reciprocal 2:1 ratio is attributed to the phosphine ligands by the hindered rotation about the CO-Ru rotation. The broad growing doublet centered at δ 38.7, with residual Ru-H coupling ($\sum$J$_{PH}$ = 15 Hz), supports the formation of the *trans*$_{P,P}$- κ$^2$-(O,O)-THAc, **4** as the more stable species (Scheme 1). The $^1$H NMR Ru-H triplets centered at −6.1 and −7.2 ppm (Figure 3), ($^2$J$_{H-Ptrans}$ = 111 Hz and $^2$J$_{H-Pcis}$ = 24 Hz) indicate that hydride substitution mainly occurred *trans* to carbonyl ligand, which is responsible for stabilizing the donor THAc ligand. The second eluted reddish fraction is the result of acetate chelation by concomitant release of PPh$_3$. THAc-chelation has confirmed by a strong Ru-carbonyl IR band at ν 1960 cm$^{-1}$ and by a narrower separation of dihapto-acetate IR bands at 1682 s, 1656 s cm$^{-1}$ (Δν = 26 cm$^{-1}$). The highfield-shifted $^1$H NMR hydride triplet at –17.0 ppm (t, $^2$J$_{HP}$ = 20.0 Hz) (Figure 3), due to the notable electron-donor dihapto ligand, is compatible with *trans*$_{P,P}$-phosphine location, analogously to the κ$^2$-O(O)-[Ru(CO)H(COH)(PCy$_3$)$_2$] complex (−18.5 ppm) [50].

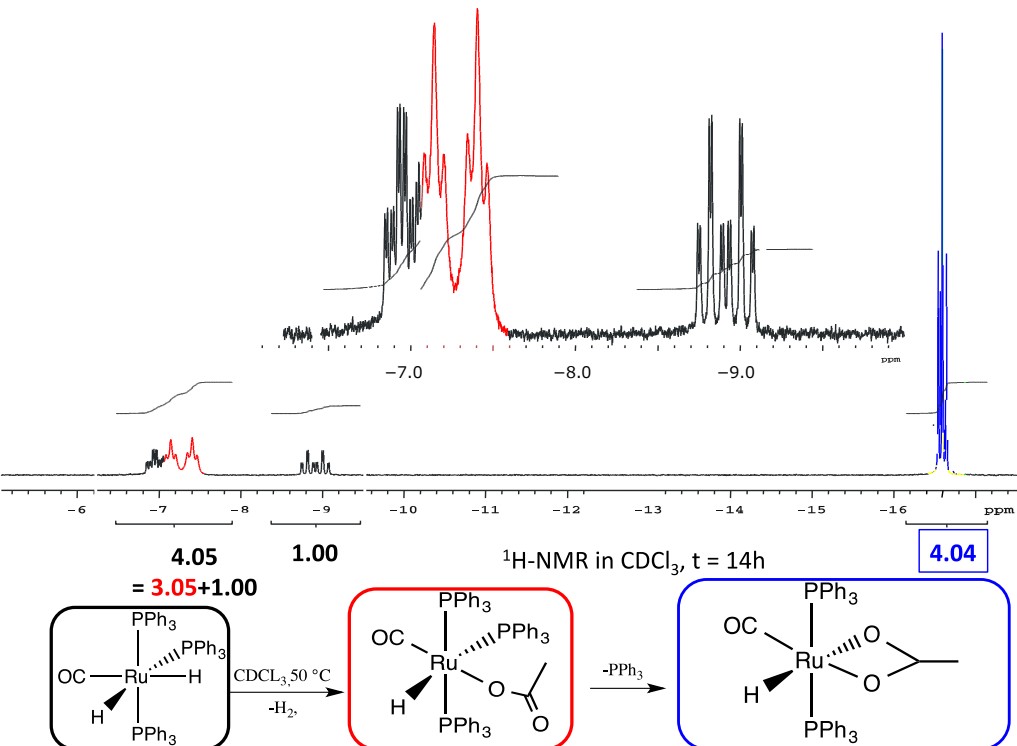

**Figure 3.** $^1$H NMR (nuclear magnetic resonance) spectrum of the two **3a, 3b** *mer*- κ$^1$(O)THAc-[Ru(CO)H (PPh$_3$)$_3$] rotamers evidences two doublets of triplets (dt, −6.12 ppm; −7.27 ppm, in red) and a triplet for the chelate *trans*$_{(P,P)}$-κ$^2$(O,O)THAc-[Ru(CO)H(PPh$_3$)$_2$] **4** (t, −17.01 ppm in blue).

Although the first reaction step, leading to κ$^1$ intermediates, was not energetically favored, the energies required were still accessible at the experimental conditions. The distinct *cis* conformers **3a, 3b** exhibited comparable energy, but were separated by a quite high barrier, as predicted by DFT calculations (Scheme 2). A third isomer, having κ$^1$-(O) *trans*-located with respect the PPh$_3$, was predicted at too high energy to be observed. The further evolution by PPh$_3$ release, forming κ$^2$-intermediates **4**, exhibited energies lower than reagents only in the case of *trans*-PPh$_3$, whereas *cis*-isomers always fall at higher energies. The last steps, leading to κ$^1$-, κ$^2$- species both in *cis* and *trans*- forms, are almost isoenergetic, as previously pointed out and fully supported by the experimental findings.

**Scheme 2.** Reaction key steps sequence to transform **1** into **2**. Monohapto intermediates **3a**, **3b** have generated by restricted C-O rotation of the THAc fragment about the phosphine ligands.

With the purpose to further investigate the reactivity path of **2**, a reaction involving subsequent addition of THAcOH was conducted under the same conditions. The addition of the first equivalent of THAc promptly formed the pale green monohapto-**3** complexes, which in turn converted the purple chelate **4**. The transformation occurred spontaneously, even in the solid state. By contrast, addition of a further equivalent of THAcH to the isolated pale green monohapto-**3** species, along the expected growing triplet at −17.0 of **4**, showed the formation of two $^1$H NMR triplets in the interval δ −10.8–11.3 ($^2J_{HH}$ = 11.7), likely attributable to transient bis-monohapto species. The related structure was suggested by DFT calculations as $\kappa^1$(O, $\kappa^2$(O)-[TH(CH$_2$)C(O)O···H···OC(O)(CH$_2$)TH][RuH(CO)(PPh$_3$)$_2$] [TH=thymine ring], in which the carboxy-carboxylic -(O)C(O)···HO(O)C- moieties of the two $\kappa^1$-functionalized arms resulted intramolecularly stabilized by a pentametallacycle, showing very similar H-bonds (1.22 vs. 1.18 A). On the energetic scale, the structure appeared almost isoenergetic (−28.1 kJmol$^{-1}$) with the trans form, used as reference, (Figure S18a). By further addition of THAc, the monohapto-, dihapto- species **2** was exclusively formed. The reaction requires one more hour at room temperature by using low-polar, non-coordinating solvents such as CHCl$_3$ to avoid further evolution to new pentacoordinate species, formed by facile releasing of phosphine ligand, as observed in DMSO.

### 3.3. DFT Theoretical Calculations

The DFT calculations, [37] performed on both the monohapto-**3** (blue line) and dihapto-**4** (red line) rotamers, indicate a high torsional barrier (50 kJ mol$^{-1}$) along the **C-O** bond for $\kappa^1$(O)-**3,** forming distinct downward or upward THAc- conformations (Scheme 2). In the case of the favorite dihapto $\kappa^2$-(O,O)-**4** species a much lower energy barrier (10 kJ mol$^{-1}$) was evaluated for the torsion along (carboxy)**C-C**(methylene) bond, confirming the equatorial THAc location as preferred for the more stable isomers, whatever coordinative mode would be adopted. Species $\kappa^1$(O)-*mer*-**3** was unstable and promptly converted to the entropically-driven chelate $\kappa^2$(O,O)-*trans*-**4**. Although not totally hindered *cis$_{P,P}$*-configurations of **3** and **4** were less favored because of steric contacts. Mixtures with different ratio of **3** and **4** intermediates were observed depending on the reaction time and the solvent nature. However, evident broadening of the $^1$H NMR triplet at −17.0 ppm, observed by keeping the solution of (**3** + **4**) in chlorinated solvent, may indicate a rearrangement by reposition of the phosphine ligands. This process seemed to be favored by the reduction of steric congestion then promoting complete isomerization. However, DFT calculations in the vacuo (Scheme 3) indicated the *cis$_{P,P}$*-(−20.7 KJmol$^{-1}$) and *trans$_{P,P}$* structures (−24.4 KJmol$^{-1}$) to be almost isoenergetic, so that rearrangement to *cis$_{P,P}$* may

result prevalently by inter-molecular H-interactions as shown by the strongly stabilized Watson–Crick H-binding contacts in the X-ray crystal packing.

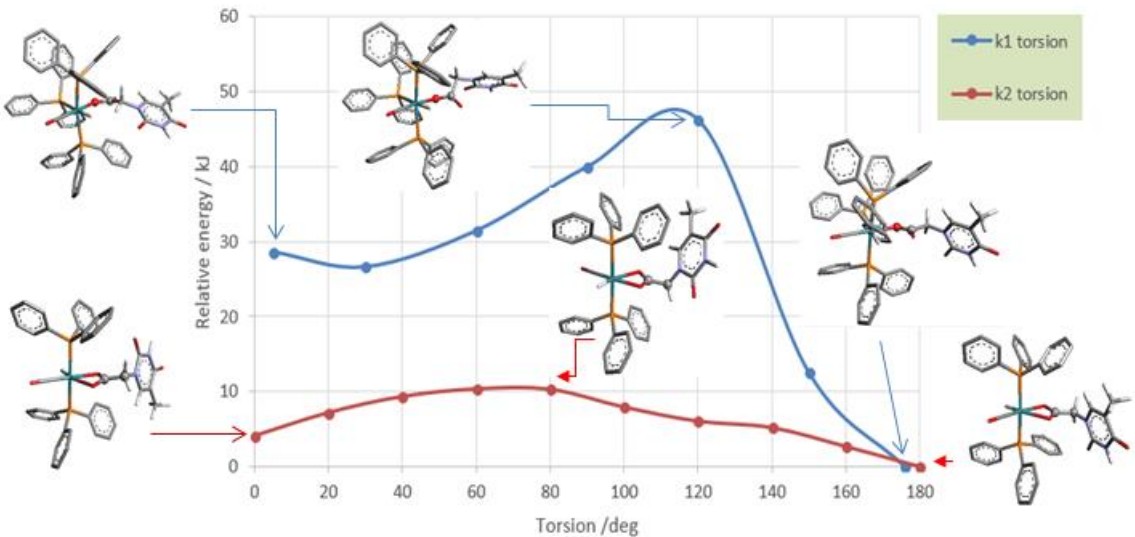

**Scheme 3.** Calculations on torsional barriers. $\kappa^1$-THAc rotation in **3** takes place along the C-O bond of the acetate group, whereas $\kappa^2$-THAc rotation in **4** is along C-C bond of the THAc-side arm. Only half of the torsion has displayed for clarity.

### 3.4. Reactions of **1** with AcOH and DFT Calculations

With the purpose to validate the proposed mechanism (Scheme 2), analogous reactions between **1** and acetic acid (AcOH) has been exploited, by changing reactant ratio, solvent polarity, and reaction time. The $^1$H-NMR spectrum of the dried solution displays two set of multiplets in 1:1.3 ratio, centered at −7.13 ppm (dt, $^2J_{PPcis}$ = 24 Hz, $^2J_{PPtrans}$ = 105 Hz) and a triplet at −16.50 ppm ($^2J_{HP}$ = 20 Hz), which are attributed to *mer*-$\kappa^1$(O)Ac-[RuH(CO)(PPh$_3$)$_3$], **5** and *trans$_{(P,P)}$*-$\kappa^2$(O,O)Ac-[RuH(CO)(PPh$_3$)$_2$], **6** respectively (Scheme 4). The selectivity results strongly dependent on the temperature and solvent polarity. The DFT-calculated energy scale (Scheme 5) displays which are the favorite species. By adding two equivalent of acetic acid, the double coordinated *trans$_{(P,P)}$*-$\kappa^1$(O)Ac, $\kappa^2$(O,O)Ac-[Ru(CO(PPh$_3$)$_2$], **7** appears as the main product. The presence of $^{13}$C NMR Ru-carbonyl triplet at 205.3 ppm, the $^{31}$P{$^1$H} singlet at δ 39.1 and the resonance at δ 0.64 for both the $^1$H NMR methyl groups, strongly support acetate interconversion at room temperature, in accord with the analogous fluxional behavior reported for [$\kappa^1$-(O)Ac, $\kappa^2$-(O,O)Ac-[Ru(CO)(PPh$_3$)$_2$] [51] by using a different reaction procedure.

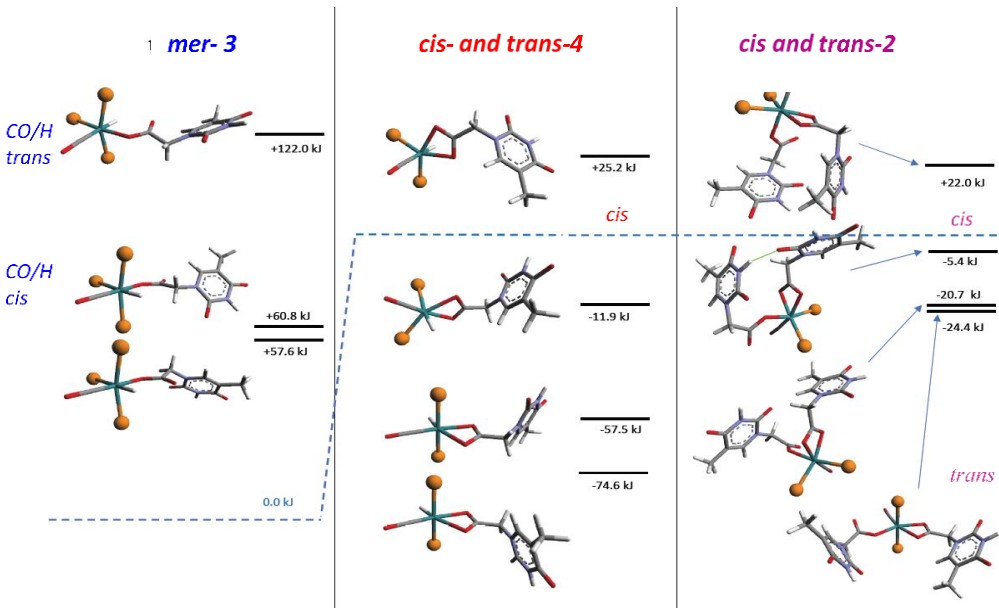

**Scheme 4.** Calculated DFT free energies (relative to the reaction of **1** with one equivalent of THAcOH (blue dash line) for the intermediates shown in Scheme 2 with other transient species intercepted spectroscopically only. It is noteworthy that the *trans* conformations are the more stable in all the cases, but the related *cis* forms are notably stabilized by intermolecular H-bonding interactions.

**Scheme 5.** Subsequent addition reactions of acetic acid (AcOH) to **1**. The reactions have been performed by adopting different conditions of temperature, time duration, reactant ratio and solvent polarity. All the optimized tests have been reported in the experimental section.

In accord with what has been described for THAcH, subsequent additions of acetic acid, prior to giving birth to the signals relative to complex **2**, unexpectedly lead to a [1]H NMR signal at $\delta$ 14.0 ppm, likely due to a novel transient penta-coordinated species (Figure S18b). The structure of the transient intermediate is assigned to a bis-monohapto species of the type $\kappa^1(O)$, $\kappa^2(O)$-[MeC(O)O···H···OC(O)Me][RuH(CO)(PPh$_3$)$_2$], showing the functionalized opened side arms interconnected by a H-bond, as suggested by DFT calculations (Scheme 6).

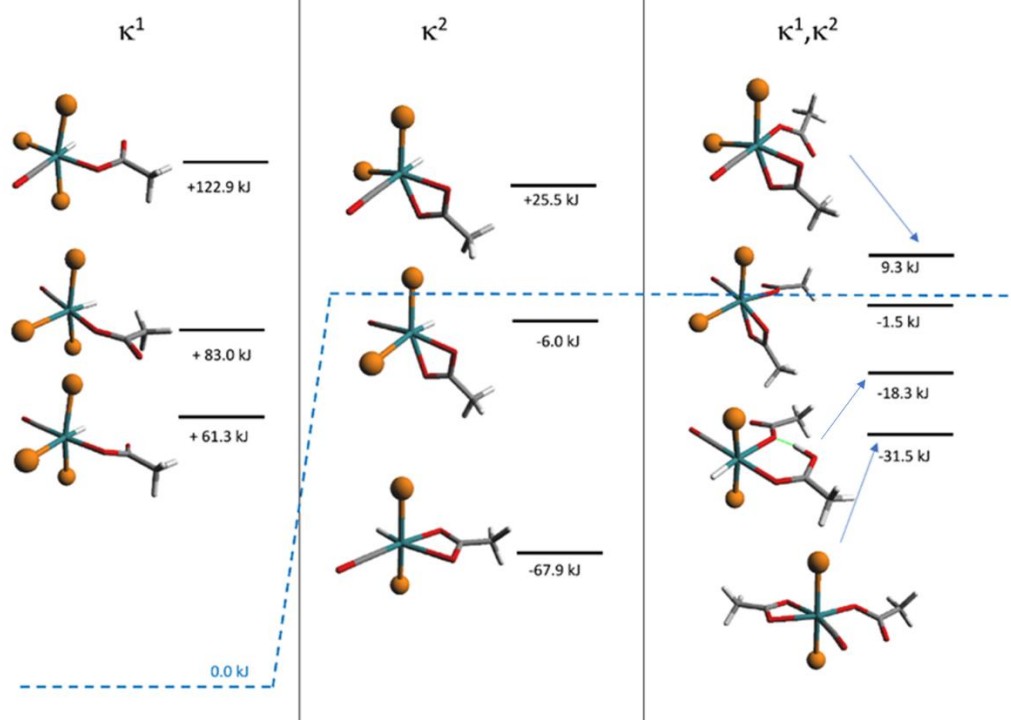

**Scheme 6.** Calculated DFT free energies (relative to **1** + **2** AcH, blue dashed line) for the reaction intermediates shown above and H-bonded postulated isomers of similar energy.

The following scheme, which demonstrates the intercepted intermediates in the sequenced reactions with AcOH, is supposed to simulate the reaction course with the THAcOH.

### 3.5. Intra or Inter-Molecular Bonding Network?

The observation of the unusually crowded [1]H NMR spectral interval in the 8–12 ppm region suggested the need to check if the supramolecular H-binding network, exhibited by the X-ray packing of **2** occurred in solution also. The idea is to attribute the signals to the corresponding structures by the help of DFT free energies, to evaluate the related stabilization scale. The DFT energies (Scheme 7) for the various THAcH keto-enol monomers indicated that the -[N(H)-C(O)]- lactam-form (**A**, also referred as *keto2* hereafter), was more stable than relative -[N=C(OH)]- iminol-lactim tautomers (**C** + **D**, referred to as *enol4* and *enol2*, respectively) [26–30]. This order does not change for THAcK salts.

The energy of (**A**) shows a remarkable decrease DG = −16.2 kJ/mol for **B**) in the case of intra-molecular H-bond interaction between carboxy proton and the CO(2).

To investigate supra-molecular H-binding networks, a model by using $\kappa^2(O,O)$-THAc$^-$K$^+$ to simulate the Ru-coordination sphere was adopted, since the full Ru-coordinated molecules were too large to be handled by computations.

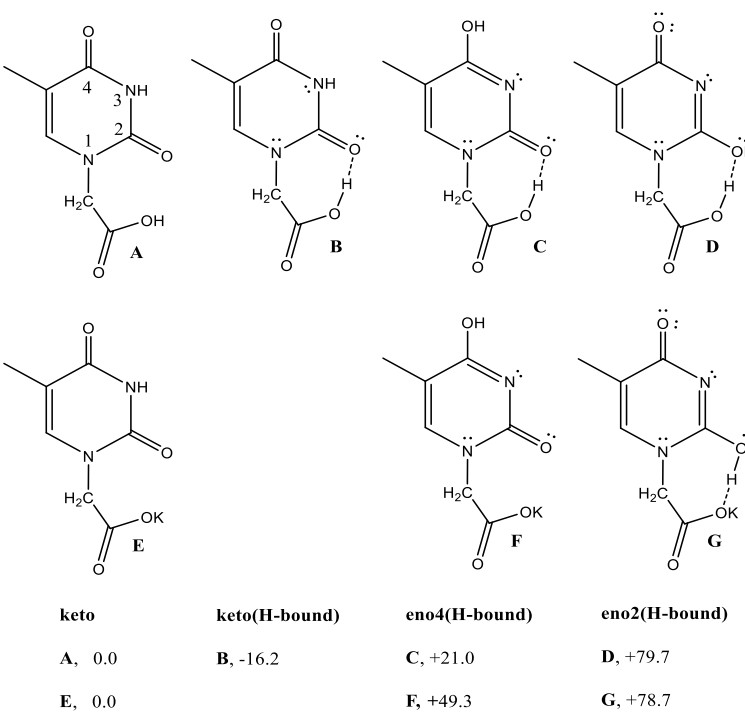

| keto | keto(H-bound) | eno4(H-bound) | eno2(H-bound) |
|------|---------------|---------------|---------------|
| **A**, 0.0 | **B**, -16.2 | **C**, +21.0 | **D**, +79.7 |
| **E**, 0.0 | | **F**, +49.3 | **G**, +78.7 |

**Scheme 7.** Calculated free energies (kJmol$^{-1}$) for the equilibrium between THAcH (**A–D**) and anionic THAcK (**E–G**) amido and imino-enol tautomers.

In spite of the bonding nature, we proposed substituting $\kappa^2$-(O,O)THAc-[Ru(CO)(PPh$_3$)$_2$] with the $\kappa^2$(O,O)-THAc$^-$K$^+$ thymine -acetate ligand, with the purpose of achieving feasible DFT calculations. In Figure 4 we compare the relative space filling models of the THAc-Ru fragment belonging to species **4** with THAc$^-$K$^+$. We chose to freeze the rotamers into two distinct perpendicular and parallel limit conformations (Figure 2). The ability of the thymine ligands to interact each other is related to the external disposition of the THAc rings with respect to the crowded volume spanned by the Ru-coordination sphere. Although the steric encumbrance of THAcK compared to $\kappa^2$(O)THAc-[RuH(CO)(PPh$_3$)$_2$] were very different, the molecular modelling evidenced that the thymine rings were external enough to reciprocally interact (Figure 4).

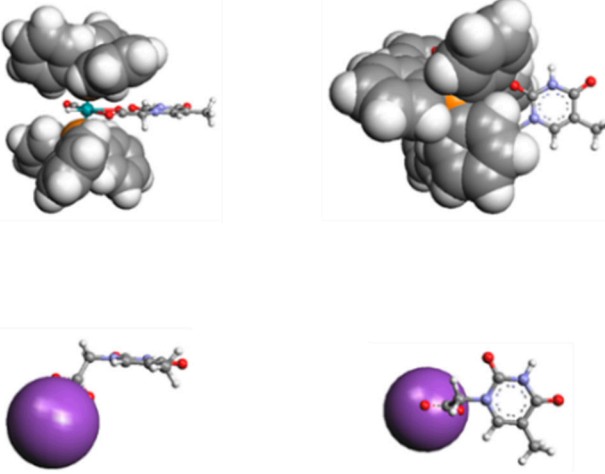

**Figure 4.** Steric comparison of the hindrance between THAcK and $\kappa^2$-(O,O)[RuH(CO)(THAc)(PPh$_3$)$_2$].

DFT calculations suggested a lot of dimerization possibilities, which are sketched in Scheme 8. All but the dimeric structures involving two enol forms (*enol4* and *enol2*), which

are found at energies larger than ~60 kJmol$^{-1}$ were not considered for the assignment in $^{1}$H-NMR spectrum.

| *c* **O2-H—O2**, *N-H—N*<br>keto2–enol2<br>53.3 | *d* **O2-H--O4**, *N-H—N*<br>keto4–enol2<br>45.9 | *e* **O4-H—O2**, *N-H—N*<br>keto2–enol4<br>41.2 | *f* **O4-H—O4**, *N-H—N*<br>keto4–enol4<br>38.4 |
|---|---|---|---|
| *g* **O4-H—O1**<br>carboxy anti–enol4<br>20.6 | *h* **O2—H-N**<br>keto2–keto2<br>18.8 | *i* **O4—H-N**<br>keto4–keto4<br>18.1 | *l* **O4—H-N**<br>keto2–keto4<br>17.4 |
| *m* **O2-H—O1**<br>enol2–enol4–carboxy anti<br>−20.4 | | *n* **O2—H-N**<br>keto2–keto2 keto4–carboxy anti<br>−31.0 | |

**Scheme 8.** Relative free energies (kJmol$^{-1}$) of THAc-K intermolecularly bound dimeric forms. The relative energies have been evaluated has half of the difference between the energy of the dimer and twice the energy of keto monomer (Structure A, Scheme 7). The O···H-N interactions are displayed in green, N-H···N in blue and O-H···O in purple and considered to have similar energy to generate coincident H-bonds in the $^{1}$H NMR spectrum. The designation of carboxy-anti has been referred to the reciprocal disposition of the C(O)O functions in the dimer. The same color notification is reported in the $^{1}$H NMR spectrum (12–7 ppm interval) (Figure 5).

We selected the space filling figure of *trans*- $\kappa^{2}$(O,O)-[RuH(CO)(PPh$_3$)$_2$], **4** as model of the (**3**+**4**) mixture solution which exhibits various $^{1}$H NMR signals in the appropriate range for H-bonding interactions. We analyzed the $^{1}$H NMR spectra (Figure 5), which contains different reaction mixtures after approximately after 45 min showing a ($\kappa^{1}$- + $\kappa^{2}$-) ratio rather equivalent ($\kappa^{1}$-**3**: $\kappa^{2}$-**4** = 1.3:1). The weak sharp $^{1}$H NMR signal at δ 11.1 was attributed to the inter-molecular anti- *enol*-carboxy dimer (entry **m**, −20.4 kJmol$^{-1}$, Scheme 8). Similarly, the three small broad signals in the interval of δ 9.2–8.8 ppm were assigned to the hydroxy function of the *enol*-lactim -N=C$_{2,4}$(OH)- species, mutually H-bound to the *keto*- tautomers (entries **c**–**f**). The $^{1}$H NMR signal at 8.3 ppm, which appears significantly more intense (5:1 with respect the C-H of thymine ring), likely belonged to the [NH-(O)C] Watson–Crick H-bonds in Scheme 6. We assumed the four overlapped N-H-N signals (7.72–7.93 ppm) to be a measure of the concomitant *enol*- species H-bound to the more stable *keto*-forms. The signals attributed to hydroxy functions were indicative for the lactim tautomers, including those involved into intramolecular H-bonding structures (entries **F**, **G** Scheme 7).

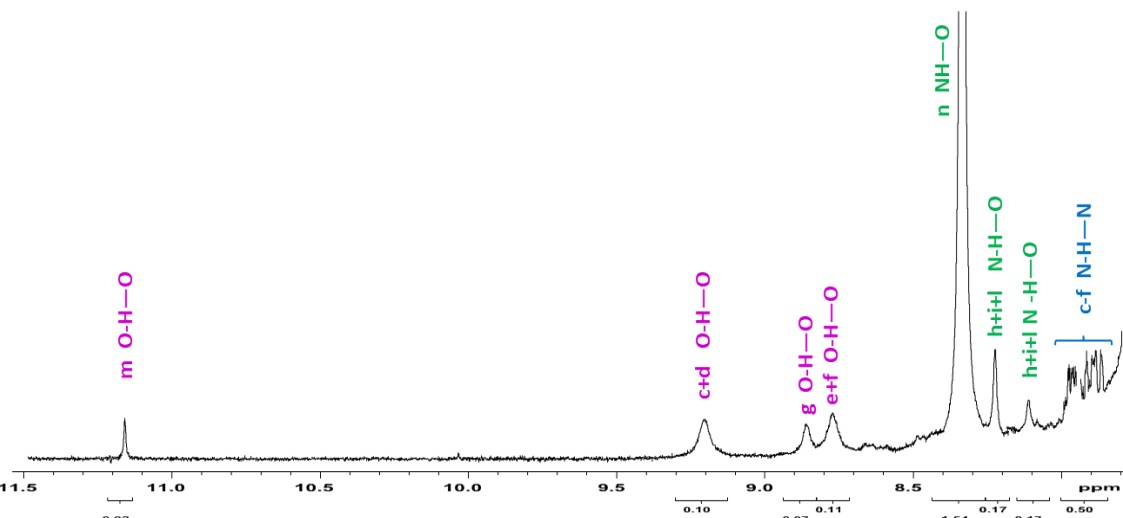

**Figure 5.** [1]H spectra relative to (**3** + **4**) mixture showing the intermediate trapping during the reaction course. The integrals evaluation for the tautomeric ratio results to be 1:3.2 with the prevalence of the stable amido conformation. The attributions of the H-bond functions of the type X–H–O (X=N, O) have been assigned based on the shielding effect (δ) O–H–O (11) < N–H–O (10-8) < N–H–N (8), related to the calculated DFT energy scale of the dimers sketched in Scheme 8.

In the case of the species $\kappa^1$(O)THAc-, $\kappa^2$(O,O)THAc-[Ru(CO)(PPh$_3$)$_2$], **2** no evidence of tautomerization has been observed in the [1]H NMR spectra (Figures S5a,b and S7a,b), since the NH-(O)C signals at δ 7.8 and 7.9 ppm indicate exclusively the formation of amino-carbonyl intermolecular bonds. To carry out this investigation we selected CDCl$_3$, which concomitantly showed low polarity (χ = 4.8) and scarce coordination ability [52]. Considering the low solubility of the observed mixture and the faint response of the X⋯H–O interactions, by comparison with the intensities of the other signal, the [1]H NMR spectra were recorded from freshly prepared samples (within 15 min) to exclude facile isotopic exchange processes with deuterated solvents. We are convinced that Ru-carboxy coordination played a pivotal role to induce proton relocation, boosting the tautomerization from -[N(H)C(O)]-amido to –[N=C(OH)]-iminol form, which finally resulted in being energetically stabilized by the inter-molecular H-binding network. The prevalence *keto* distribution (3.2:1) is correlated to inter-connections shown by the dimeric species, proposed as case-study model, and reported in Scheme 7. The NMR signal attribution to the X-H-O (X=N, O) intermolecular interactions was based on the electron withdrawing ability, the shielding effects of O–H–O (**c–g, m and intramolecularly G**) < N–H–O (**h–i, n**) < N–H–N (**concomitant c–f**) chemical shifts scale, the DFT-calculated energy scale and correlated to the resonance intensities. The carboxy-*enol* signal interaction (δ or 11.2) and the minor hydroxy signals around δ 8, which all belong to the minor *enol*-forms, were compared to the intense resonance at δ 8.3 which represented the prevalent *keto* isomers. The presence of iminol forms was also responsible for the concomitant growing of the NH-N signals (δ 7.8–8.0). As suggested by DFT-space filling evaluations, the H-bond interactions have merely been considered in the case of thymine rings *anti*-disposed to minimize the steric interference with the crowded *trans*-phosphines, therefore allowing the remarkable H-bond network. (Figure 5). A variety of [1]H NMR spectra of intermediate mixtures, because they were analyzed at different reaction times, and although composed by different proportion of chelate-**4** and monohapto-**3** structures, showed a similar low-shifted signal pattern (two distinct examples are reported in Figure S11b,c). Considering the uncertainty of 10–15% due to integral determination and the different proportion of $\kappa^1/\kappa^2$ coordination of the non-conjugated metal fragments, the analyzed spectra exhibited an *enol/keto* ratio of 0.31 (Figure 5), compared to the enol/keto ratio as 0.13, reported in the literature [53].

The assignments were accomplished by the help of DFT-calculated energies for THAc anionic dimers selected as models, supporting the belief that the intermolecular interac-

tions, promoted by H-binding network, played a key role in the solution also. The larger line widths, which were observed for the intermolecular O···H···O bonds (violet entries **c–f**) may have been due to the higher degree of freedom shown by OH···O(C) functions, which presented two O-lone pair able to generate in turn more than one H-bonded conformation. On the contrary, NH···O bond type (green entries **n–l and n**) or the enol-carboxy C(O)O···HO interactions at δ 11. 1 (**m**), which can occur intra- (**G** entry in Scheme 7) or intra-molecularly (entries **h–l and n** (Figure 5), resulted in being sharper since they are affected by a stronger H-bond strings network as in the case of N–H–N interactions (blue entries **c–f**).

## 4. Conclusions

The report deals with the reaction of **1** with THAcH excess, forming double coordinated mononuclear $\kappa^1$(O)THAc-, $\kappa^2$(O,O)THAc-[Ru(CO)(PPh$_3$)$_2$] species **2**, which indicates the versatility of the coordination modes exhibited by thymine acetate. However, the X-ray characterized *cis$_{P,P}$-fac$_{THAc}$* isomeric form of **2** did not correspond to the *trans$_{P,P}$-mer$_{THAc}$* structure observed in solution. A single $^1$H NMR signal for the THAc methyl moieties indicated a low-energy monohapto–dihapto interconversion at room temperature. DFT energetic studies confirmed the stabilization in solution of the *cis* configuration, governed by the reduced steric hindrance and by maximizing either intra- and inter-molecular H-bonding or π–interactions. Metal hydride upfield-shifted $^1$H NMR signals indicated the nature of the isolated intermediates **3** and **4,** the rotamer energies of which were studied also by DFT-calculations. To elucidate the reaction course, similar reactions were run between **1** and acetic acid, confirming the proposed mechanism of *cis-trans* interconversion by the help of DFT calculations, despite the remarkable reduced steric requirements and the limited H-binding features. Upon ruthenium complexation, due to the further stabilization imparted by X–H–O (X=O, N) supramolecular H-binding network, the rare -(O)C-N=C(OH)- THAc-lactim tautomer results increased remarkably (0.31) compared to the predominant (O)C-NH-C(O) lactam form.

Lippert's pioneering work, by dealing with the mutagenic properties of metallodrugs in anticancer activity, suggested proton relocation was responsible for the stabilization of rare tautomeric forms, being promoted by direct (N,O) metal coordination of the thyminate nucleobase. Herein, we further support the evidence that even non-conjugated thymine derivatives, once coordinated to ruthenium, may notably influence the stability of the minor iminol (0.31 ratio) compared to the tautomeric thymine equilibrium (pka = 10), where the formation of iminol species is promoted by acidic treatment. The occurrence of the latter tautomer, further stabilized by the preponderant H-binding network, is recognized to be active in DNA mismatching, limiting the replication of cancer cells [54–57].

**Supplementary Materials:** The following are available online at https://www.mdpi.com/article/10.3390/app11073113/s1, Table S1: Crystal data and structure refinement for compound *cis*-**2**, Table S2: Most relevant hydrogen bonds for *cis*-**2** [Å and °]; Figure S1: Molecular structure of *cis*-**2** with the atom labelling, Figure S2: Molecular structure of *cis*-**2** with π-π stacking and Watson-Creek intermolecular interaction, Figure S3: View down. Fragment of the crystal packing of complex **2** illustrating the intermolecular H bonding pattern. For the sake of clarity only the H atoms engaged in H bonds are shown. Ball and stick representation is used for the dimer arising by the strong N-H···O hydrogen bonding and for the atoms connected to it. H bonds are shown with blue dashed lines. The *a* axis of the crystal packing of *cis*-**2**. Black dotted lines indicate the intermolecular N-H···O hydrogen bonds, Figure S4: Fourier transform infrared (FT-IR) spectra of trans-$\kappa^1$(O)THAc-, $\kappa^2$(O,O)THAc-[Ru(CO)(PPh$_3$)$_2$] **2**, Figure S5a: $^1$H-NMR (400 MHz) of cis-$\kappa^1$(O)THAc-, $\kappa^2$(O,O)THAc-[Ru(CO)(PPh$_3$)$_2$] **2** in CDCl$_3$, Figure S5b: $^1$H-NMR (400 MHz) of cis-$\kappa^1$(O)THAc-, $\kappa^2$(O,O)THAc-[Ru(CO)(PPh$_3$)$_2$] **2** full spectrum in CDCl$_3$, Figure S6: ESI-MS of cis-$\kappa^1$(O)THAc-, $\kappa^2$(O,O)THAc-[Ru(CO)(PPh$_3$)$_2$] **2**, Figure S7a: $^1$H-NMR (400 MHz) of cis-$\kappa^1$(O)THAc-, $\kappa^2$(O,O)THAc-[Ru(CO)(PPh$_3$)$_2$] **2** in CDCl$_3$, Figure S7b: $^1$H-NMR (400 MHz) of cis-$\kappa^1$(O)THAc-, $\kappa^2$(O,O)THAc-[Ru(CO)(PPh$_3$)$_2$] **2** aromatic region in CDCl$_3$, Figure S8: FT-IR (KBr) $\kappa^1$(O)THAc-[RuH(CO)(PPh$_3$)$_3$] **3a, b**, Figure S9: $^{31}$P{$^1$H} NMR (161 MHz) of **2** in CDCl$_3$, Figure S10: FT-IR (KBr) of $\kappa^2$(O,O)THAc-

[RuH(CO)(PPh$_3$)$_2$] **4**, Figure S11a: $^{31}$P{$^1$H} NMR (161,9 MHz) κ$^1$(O)-[RuH(CO)(THAc)(PPh3)3 **3a,b** + κ$^2$(O,O)-[RuH(CO)(THAc)(PPh$_3$)$_2$] **4** in CDCl$_3$, Figure S11b: $^1$H-NMR spectrum of **3+4** (ratio 3:2) showing iminol/keto ratio = 0.31 in CDCl$_3$, Figure S11c: $^1$H-NMR spectrum of **3+4** (ratio 3:7) in CDCl$_3$ showing iminol/keto ratio = 0.25, Figure S11d: Complexes formed as impurities during the preparative reduction of starting material **1**, Figure S12: Electrospray ionization mass spectrometry (ESI MS) (MeOH) of κ$^2$(O,O)THAc-[Ru(CO)H(PPh$_3$)$_2$] **4**, Figure S13: $^1$H-NMR hydride region spectrum of **1**+ κ$^1$(O)Ac-[RuH(CO)(PPh$_3$)$_2$] **5**+ κ$^2$(O,O)Ac-[RuH(CO)(PPh$_3$)$_2$] **6** in CDCl$_3$, Figure S14a: FT-IR of compound **5**, Figure S14b: $^{31}$P{$^1$H}-NMR spectrum of κ$^2$(O,O)Ac-[RuH(CO)(PPh$_3$)$_2$] **6** species, Figure S15: $^{13}$C{$^1$H}-NMR spectrum of **6** species, Figure S16: FT-IR (KBr) κ$^1$(O)Ac-, κ$^2$(O,O)Ac- [Ru(CO)(PPh$_3$)$_2$], **7**, Figure S17: Carbonyl region in $^{13}$C{$^1$H}-NMR spectrum of (*trans+cis*)(P,P)- κ$^1$(O)Ac,- κ$^2$(O,O)Ac-[Ru(CO)(PPh$_3$)$_2$] **7** in CDCl$_3$, Figure S18a: portion of $^1$HNMR spectrum in hydride region of intermediate κ$^1$(O)-, κ$^2$(O)-[THCH$_2$C(O)O⋯H⋯OC(O)CH$_2$TH][RuH(CO)(PPh$_3$)$_2$][TH=thymine ring] **A**, Figure S18b: portion of $^1$HNMR spectrum in hydride region of intermediate κ$^1$(O), κ$^2$(O)-[MeC(O)O⋯H⋯OC(O)Me][RuH(CO)(PPh$_3$)$_2$] **B**, Figure S18c: $^1$H-NMR portion of the spectrum of **A** in CDCl$_3$ displaying the CH$_2$ moiety of the Et$_2$O solvent, remarkably shifted upon Ru-coordination, Figure S18d: $^1$H-NMR portion of the spectrum of **A** in CDCl$_3$ displaying the CH$_3$ moiety of the Et$_2$O solvent, shifted upon Ru-coordination

**Author Contributions:** Conceptualization, S.B.; methodology, S.C. and G.M.; software, R.T. and M.M.; formal analysis, S.C., M.M., C.B. and R.T.; investigation, S.B., S.C. and G.M.; resources, S.B.; data curation, S.C. and C.B.; writing—original draft preparation, S.B.; funding acquisition, S.B. All authors have read and agreed to the published version of the manuscript.

**Funding:** This research received no external funding. We thank the University of Bologna.

**Institutional Review Board Statement:** Not applicable.

**Informed Consent Statement:** Not applicable.

**Acknowledgments:** We thank the University of Bologna. Pietro Paolo Cristallini, Fabio Battaglia and Federica Zizzi are also gratefully thanked for the relevant and accurate experimental work carried out during their graduation internship.

**Conflicts of Interest:** The authors declare no conflict of interest.

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
