# Peer review of "Ruthenium–Thymine Acetate Binding Modes: Experimental and Theoretical Studies"

_applsci, doi:10.3390/app11073113_

Round 1
Reviewer 1 Report
Opinion on the paper „Ruthenium–Thymine Acetate binding modes: Experimental and Theoretical studies”
The manuscript is on the reaction of thymine acetate with ruthenium complex. The reaction mimics the interaction of ruthenium ions with DNA. The authors obtained a ruthenium compound in which there are two different ways of coordinating the metal center. Based on the structure of the isolated intermediate products, a mechanism for changing one isomer into another has been proposed. With the purpose to validate the proposed mechanism, an analogous reaction between starting ruthenium complex and acetic acid has been studied. Eexperimental results were supported by theoretical calculations of the transient states. Structure of the compounds were determined by X-ray crystallography, and NMR and IR spectroscopy.
Overall, this is a well written manuscript. This work is novel. The experimental techniques which have been used in this work are appropriate and the data of this work are reliable and sufficient. After minor revision the paper should be published in the Applied Sciences.
Commends:
- Page 2, line 48 [3,16], References should be cited sequentially. Instead of [3.16] it should be [3.6].
- Page 2, line 67, Reference is needed.
- Pages 15 and 16, References should be arranged in accordance with the publisher's instructions.

Author Response
I did all the suggested alterations
Reviewer 2 Report
The article describes broad study of ruthenium conjugate. Those include synthetic optimization, X-ray crystallography, NMR studies, DFT calculations, and mechanistic discussion. Thus, it is one of the highly comprehensive study presented. The Ru complex forms Watson-Crick pair in a solid state, as determined by crystallography. This study expands limited so far knowledge of interactions of ruthenium conjugates, and is nicely written.
I would welcome this article published in Applied Sciences pending very minor revision addressing issues outlined below:
- Page 2 line 67, please give reference to the method of synthesis of compound 1.
Author Response
I did all the suggested alterations
Reviewer 3 Report
See attached file

Author Response
I polished the English and correct misspelling words and phrases to my best. I changed the sentences as suggested and I added ref, if missed.
I explain the dissymmetry related to the trans-complexes for justify the unexpected mutual coupling between the anisochronous trans-phosphine ligands
Page 3 line 132
I included the 31P spectrum in the SM
I correct the picture and the number sequence f the Figures and Schemes, apologizing for the distraction
I re-writed the bibliographic section
Unfortunately both the compounds 2 show remarkable instability and they are prone to prompt transformation in solution:
Trans- compound converts to cis and the cis form has readily transformed into pentacoordinated structure, which decomposed by interaction even with weak coordinating solvents. One of the crystals examined by X-ray reveled to be coordinated by CH2Cl2 or EtOH, that has been used for dissolving the powdered solid for further purification procedure. I ask colleagues to examine compound 3 and 4 by solid state NMR, sending the powdered samples by mail, but their nature has changed: For instance the monohapto 3 was transformed into the chelate 4 by releasing of (Ph3)3PO.
Reviewer 4 Report
The paper presented as Manuscript ID: applsci-1123231 deals with a mode of coordination of Thymine Acetic Acid (THAcH) selected as a model for nucleobase derivatives in respect to dihydride ruthenium as a mer-[Ru(H)2(CO)(PPh3)3] compound. The ligand is coordinated through the side arm carboxylate group. Several ruthenium compounds were synthesized and characterized by spectroscopic methods, such as: IR, 1H-, 13C- 31P-NMR. The structure of one of the compounds has been studied by single crystal X-ray analysis, namely the cis-P,P-fac THAc-isomer of k1(O)-THAc, k2(O,O)THAc[Ru(CO)(PPh3)2], where the two THAcH ligands are in a reciprocal facial disposition one being monodatate (monohapto) and the other being bidantate (dihapto) coordinated. It was confirmed that the formation of this solid state attractive supramolecular architecture has been at the expense of inter- and intra- H-bonding and and p-interactions. It’s corresponding trans-isomer was observed in solution. Using 1H- NMR and DFT calculations it was found that the conversion from monodentate to bidentate THAc coordination in solution has a low energy barrier. The cis-trans interconversion was investigated by a model reaction using a simpler ligand acetic acid. It was suggested that the tautomeric THAc lactam – lactim equilibrium upon ruthenium coordination was influenced by supramolecular H-bonding network and the main result is stabilizing the minor [(N=C(OH)] lactim tautomers, that is important for antitumor activity . The main conclusion that “the relocation of protons in the Ru-coordinated nucleobase thymine derivatives enhances the occurrence of the minor iminol forms which may be active in DNA mismatching by limiting replication of cancer cells” deserves special attention.
I’ll give several remarks in order to improve the article:
- Introduction: It needs to be improved by providing additional data on the benefits of ruthenium as a basis for anticancer drug design and most importantly to explain Lipperts' proposal for the presence of minor iminol forms of Ru-coordinated nucleobase derivatives of thymine that were stabilized by proton displacement.
- The oxidation state of ruthenium should be discussed as well, for example it is interesting why the 1H-NMR signals, in the spectrum presented on Fig. 5 are to much wide.
- The sentence “asymmetric larger absorptions for acetate at 1653 and 1363 (Δν=290 cm-1) “ should be“antisymmetric and symmetric stretching vibrations of the COO- -group of acetate at 1653 and 1363 (Δν=290 cm-1) “
- It would be better to apply DFT calculations not only to estimate the energy but also to obtain simulated NMR spectra, especially when interpreting the NMR spectra of mixtures.
- I cannot judge the single-crystal X-ray solution of the cis-isomer 2, as so far I have no information from the CCDC about the structure No. 1868289, but in any case the positions of the atoms in Fig. 1, must be represented by thermal ellipsoids
- There are many incorrect designations – for example Rows. No 194. 195: it should be “trans-P,P-configuration” or “merTHAc stereogeometry
- Row 478 – may be a trans – isomer in solution or?
Author Response
I’m truly grateful to referee 4 for the comments which invite me to explain in detail all the results I did not present in the first manuscript draft for clarifying some points. I prefer to directly add all the informations you asked in the revised manuscript, since are all supporting and clarifying the explanations on the carried out experiments.
I polished the English and correct misspelling words and phrases to my best. I changed the sentences as suggested and I added ref, if missed.
I explain the dissymmetry related to the trans-complexes for justify the unexpected mutual coupling between the unsynchronous trans-phosphine ligands
Page 3 line 132
I included the 31P spectrum in the SM
I correct the picture and the number sequence f the Figures and Schemes, apologizing for the distraction
I re-writed the bibliographic section
Unfortunately both the compounds 2 show remarkable instability and they are prone to prompt transformation in solution:
Trans- compound converts to cis and the cis form has readily transformed into pentacoordinated structure, which decomposed by interaction even with weak coordinating solvents. One of the crystals examined by X-ray reveled to be coordinated by CH2Cl2 or EtOH, that has been used for dissolving the powdered solid for further purification procedure. I ask colleagues to examine compound 3 and 4 by solid state NMR, sending the powdered samples by mail, but their nature has changed: For instance the monohapto 3 was transformed into the chelate 4 by releasing of (Ph3)3PO.
- The introduction has been changed following the given suggestions
- The oxidation is always (II) and it would not change under the mild reaction conditions utilized. Full spectra are added in Supplementary Materials. The width of the signals is therefore discussed in the text.
- Done
- We made some attempts to predict the NMR spectra fully ab initio but, due to the size of the species, we had to use a rather low level of theory. Thus, the predicted spectra resulted to be not accurate enough to assist the interpretation of experimental spectra.
- We have checked the correspondence between the CCDC- 1868289 number in the present paper and the CCDC number for the structure of cis-2 deposited in the Cambridge Crystallographic Data and found out that there is no mistake. For refereeing purposes it is always possible as a referee to retrieve the .cif file from the CCDC . One must simply fill in a form at ccdc.cam.ac.uk with the personal details and the requested CCDC number . The service is very efficient since in less of half an hour one receives by E-mail the .cif file related to an unpublished paper. It is also useful for retrieving cif files of crystal structures not yet available in the CCDC but already present in published papers
- Done
- Done
Round 2
Reviewer 3 Report
See attached file

Author Response
Dear Referee 3, thank you for your detailed revisions and please see the attached file
Many regards Silvia Bordoni

Round 3
Reviewer 3 Report
As stated in my previous reviews of the manuscript, the work by Bordoni and coworkers describes experimental and theoretical studies regarding the binding modes between ruthenium and thymine-1-acetate (THAc) in order to clarify the structure-activity relationship. In its current version, the authors have clarified most of my concerns. Therefore, the results pointed out in this manuscript are amenable for publication at Applied Science. Nevertheless, minor correction is still required.
English should be polished all along the manuscript.
Results-Some typographical mistakes found
- Page 6. Scheme 1. Scheme 1 is incomplete. The two top figures have been cut when pasted into the manuscript. Please, correct it.
Would it be possible to include your information in the answer to referee 3 letter regarding the nature of the intermediates, including the scheme, as part of the supplementary information?
Author Response
I added every point suggested by reviewer 3 into the main text, which has been evidenced in red to distinguish from the previous one
I substituted the scheme 5 with one evidencing the trapped intemediates in the manscript
I added the cis form 7b in the Scheme 5 since it is clarifiying the mechanistic profile in analogous reaction path with AcOH, as well as the relative spectroscopic values in the experimentals
I added two more spectra in the Supplementary Material